# Traffic Safety at German Roundabouts—A Replication Study

Andreas Leich [1,*,†] , Julian Fuchs [1,†], Gurucharan Srinivas [1,†], Joshua Niemeijer [1,†] and Peter Wagner [1,2,†]

1  Deutsches Zentrum für Luft- und Raumfahrt e.V., Institute of Transportation Systems, Rutherfordstrasse 2, 12489 Berlin, Germany; j.fuchs@dlr.de (J.F.); gurucharan.srinivas@dlr.de (G.S.); joshua.niemeijer@dlr.de (J.N.); peter.wagner@dlr.de (P.W.)
2  Institute of Land- and Sea Transport Systems, Technical University of Berlin, Salzufer 17-19, 10587 Berlin, Germany
*  Correspondence: andreas.leich@dlr.de; Tel.: +49-30-67055-409
†  These authors contributed equally to this work.

**Abstract:** Roundabouts are well-known for their ability to improve upon traffic safety, especially for motorized traffic. An in-depth analysis on this topic is known from previous work. It was found that different types of roundabouts have different levels of safety. The work at hand is a replication study for a previous study in this regard. It uses a mix of traditional and a Machine Learning (ML)-based approach, expands on the previous results and replicates some of the previous findings. This was possible especially by using a factor of 10 more roundabouts in the analysis, with considerably less manual intervention. Furthermore, this study could also draw some additional conclusions regarding the safety of bicyclists, which were not included in the original study. Finally, by using cross-validation techniques, a kind of minimal model could be established that needs fewer factors and achieves better prediction quality than straightforward glm models.

**Keywords:** road safety; crash rates; traffic safety; roundabouts; machine learning

## 1. Introduction

Roundabouts are well-known for their ability to increase traffic safety at a particular kind of intersection. The results from a meta-analysis [1] (see also the older [2]) indicate that the conversion to a roundabout decreases the number of fatalities by 65%, and the number of crashes with injuries by 40% (see also [3]). The effect on property damage, however, was unclear.

This amounts to motorized traffic; for vulnerable road-users the picture is less bright. e.g., the study [4] found:

> "Vulnerable road users are more frequently than expected involved in crashes at roundabouts and roundabouts with cycle lanes are clearly performing worse than roundabouts with cycle paths."

As with other road traffic infrastructure, roundabouts differ in dangerousness related to their complexity; the more arms there are, the higher the crash-rate, and the same holds for multi-lane roundabouts [3–6].

To dive deeper into the details of what renders a roundabout safe or dangerous, Ref. [7] (in German) performed a detailed study where they classified German roundabouts into four distinct classes named A, B1, B2, and B3. Please see Section 2 for more details on this classification. Altogether 100 roundabouts were included in their study; they were carefully selected and each of them analysed in painstaking detail. They analyzed a wide range of factors that may or may not change the dangerousness of a roundabout. Their main result was that the crash-rate of the four classes differed considerably, with type A being the safest one. These results are displayed in Figure 1; however, it is not completely clear which of the differences between the crash-rates displayed are statistically significant.

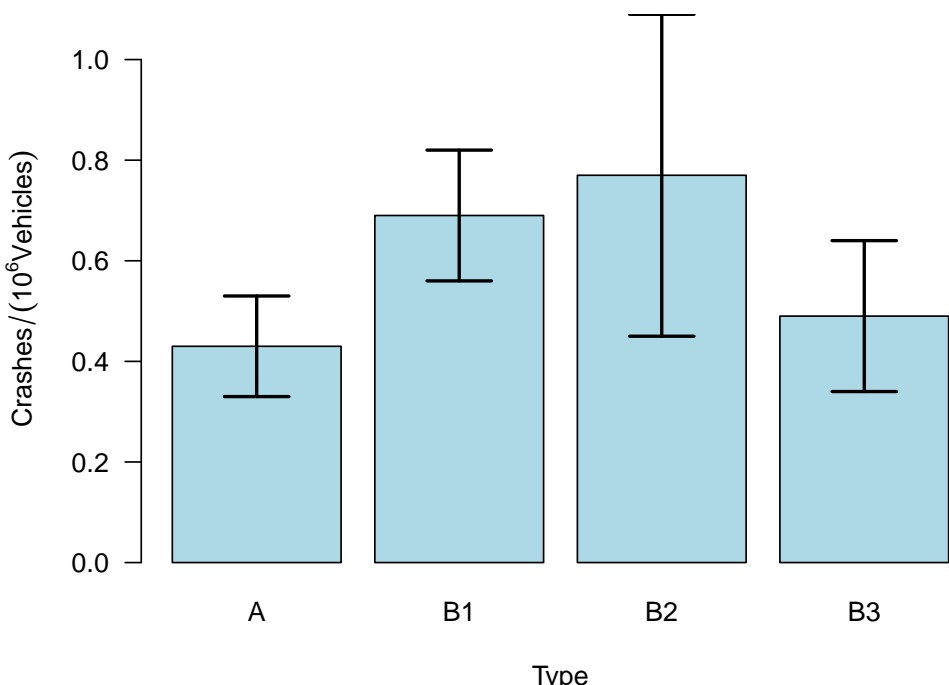

**Figure 1.** Crash-rate for the four different types of roundabouts defined by [7]. The error bars are the confidence intervals. The data for this plot have been taken from their report.

## 2. Methods

Instead of conducting a detailed study of each roundabout, this work tries a less laborious approach. A number of databases to be detailed below have been used and fused together to find a richer dataset, but eventually one that was not as carefully selected as the one in [7]. Only data from the German federal state Northrhine-Westfalia (NRW) were available and used, but the general methodology is transferable to other study areas as well.

### 2.1. Types of Roundabouts and Manual Post Processing of Data

This replication study follows the classification of roundabouts in [7]:

- Type A: Bicycles are supposed to travel in mixed traffic with motorized road users. They *do not* have a separated bike path.
- Type B1: Bicycles have their own dedicated bicycle ford for crossing the road painted on the pavement; this means, especially, that they *have the right-of-way* at the crossings of the paths of motorized and bike traffic and that they *do* have own separated cycle paths;
- Type B2: Bicycles have a common bicycle ford that they share with pedestrians and *do* have cycle paths. As with type B1 they *have the right-of-way* when crossing the path of motorized traffic;
- Type B3: Bicycles *do* have separate bike paths next to the main road and must *yield the right-of-way* to motorized road users.

All the roundabouts considered in this study are single lane roundabouts with three, four, or five access roads. The majority of the roundabouts had four access roads. Since Bondzio et al. did not see any statistically significant influence of the number of access roads on the accident rates, the number of access roads was not included in the data record. For the same reason, the diameter of the roundabout was also not included.

It is sometimes difficult to differentiate between types A and B2. In such cases, the recognized traffic signs and lane markings in street view and aerial images are of great help: roundabouts without a dedicated bicycle ford but with a zebra crossing are of type

B2 if there is a traffic sign stating that bicycles may use (see Figure 2) the sidewalk. If the German traffic sign "Z 241" or "Z 240" is present or a lane marking for a bicycle path on the sidewalk could be observed, the type was set to B2 (see Figure 3). It was classified as type A otherwise.

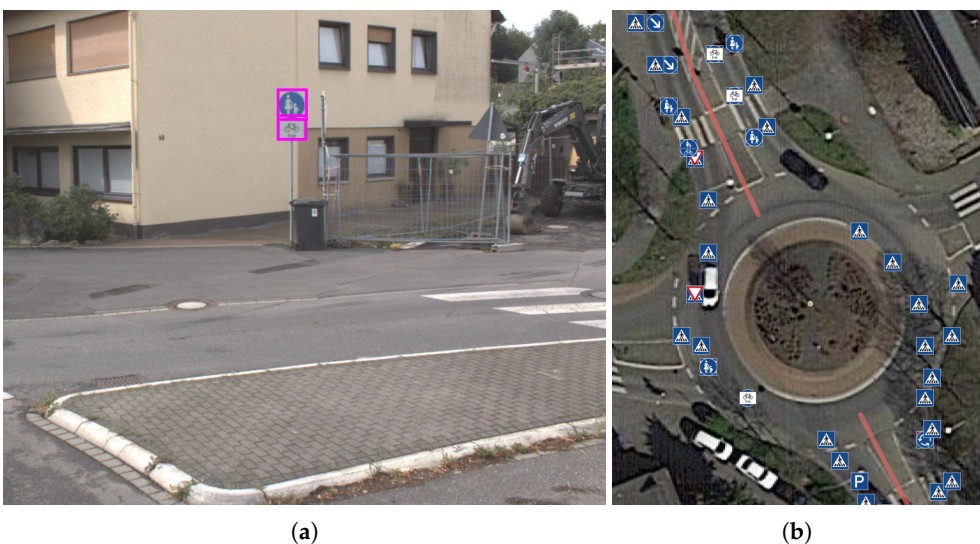

(**a**)                                                                                (**b**)

**Figure 2.** Differentiating Type A and B2 roundabouts using Street View (**a**) and Top View (**b**) in QGIS.

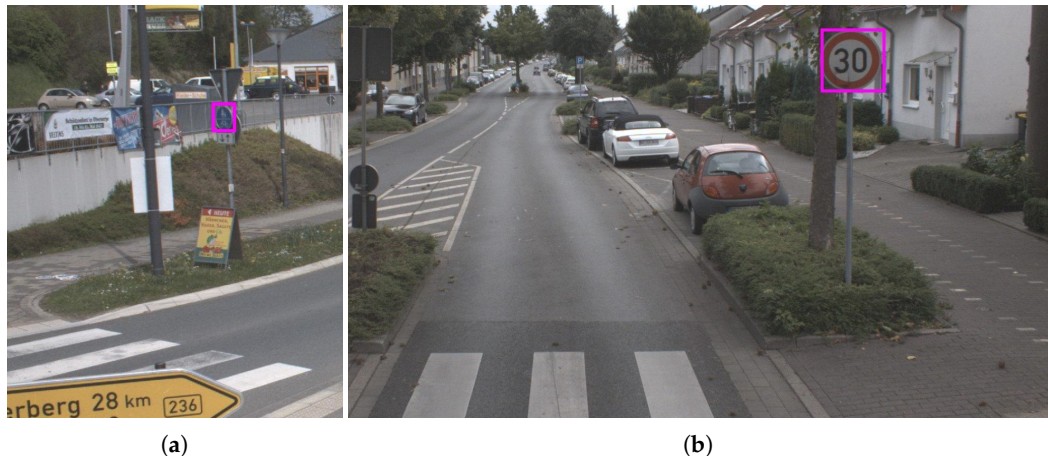

(**a**)                                                                                (**b**)

**Figure 3.** Differentiating Type A and B2: presence of traffic sign "Z 240" (**a**) and presence of a bicycle way (**b**).

The deep learning algorithms described in Section 2.3 were used to analyse the street view data from the periodic inventory of the NRW roads data collection. The information about the presence of traffic sign detections was added to the geo-database in the Open Source Geographic Information System QGIS ([8]) and therefore was available for browsing, filtering and searching. This allowed the data processing to be as efficient as necessary to handle the large number of data entries in the sample.

### 2.2. Data Aquisition

The following databases have been used:

- The road database of NRW, called NWSIB [9]. Here, especially street view images from the periodic data collection of the larger roads in NRW were used. The ML (see Section 2.3 for more details) was trained to pick the traffic signs from these pictures which enabled us to identify (among other signs) the roundabouts;

- This road database contained for most (but not all) roundabouts the information about the average daily traffic (ADT); in the best of all cases, they were divided into car counts and bike counts. Again, the simplest approach was chosen and only the ADT-values within a radius of 75 m around each roundabout were selected;
- The crash database of Germany [10], which is publicly available. As with the ADT values, each crash within a certain radius of 75 m around the roundabout was used in the subsequent analysis. Note that this approach is very different from the one in [7], where they acquired for each of the roundabouts all the detailed crash-reports and explicitly checked that it "belonged" to the roundabout.

All this resulted in a set of about 1300 roundabouts from this federal state alone. The classification into A, B1–B3, however, was still performed manually, since there was neither the opportunity nor the resources to train yet another ML model to do this classification. In addition, doing this manually (see Section 2.1) served as an additional check of the data selected, and it eliminated a few percent of the roundabouts. So, this approach ended up with a consolidated list that was used for the analysis below.

Further, the overlap between the 1300 roundabouts and the 100 roundabouts from the year 2011 considered by [7] was determined. This yields an impression about the quality of the data.

Notes on the current sample of 1300 roundabouts:

- It is beyond the scope of the current analysis whether the roundabouts were designed and constructed according to the German guidelines. However, it is assumed that this is the case, because road construction is well regulated in Germany;
- The sample may or may not include roundabouts that were being road safety audited or treated by the road accident commission. This is in line with [7], but may also be a shortcoming. Supposed road safety auditors or road accident commissions frequently add red painted bicycle fords and zebra crossings to a roundabout that is dangerous for other reasons, and that could introduce a bias that is unrecognized in this study;
- As usual, roundabouts in urban areas in Germany have a speed limit of 50 kph. Roundabouts outside urban areas may have a speed limit of 70 kph, 80 kph, or even 100 kph. This information is not covered in the current dataset. However, for roundabouts in urban areas, whether at least one of the access roads had a 30 kph speed limit (kph30an) or the whole roundabout was within a 30 kph speed limit zone (kph30in) was recorded.
- The functional classification of the roads is not included in the dataset. In Germany, there exist the three types of roads, RQ 7.5 (5.5 m standard road cross section, ADT < 3000 Veh./day, usually district roads), RQ 9.5 (6.5 m standard road cross section, ADT < 15,000 Veh./day, usually state roads), and RQ 10.5 (7.5 m standard road cross section, ADT < 20,000 Veh./day, usually federal roads). Roads of all types are present. The functional classification, however, is included implicitly because the ADT values are included in the record of each roundabout.

### 2.3. Data Mining Using Machine Learning

Machine learning (ML) was used to recognize traffic signs, which was very helpful in finding the traffic circles and relevant features to classify them in the NRW road database. This section contains a sketchy description of what was done.

"Faster-region based convolutional neural network" (Faster-RCNN) (see [11]) was employed for detecting relevant traffic signs in street view images. Faster-RCNN is more suitable compared to non-region-based detectors when dealing with objects that are relatively small in the image (e.g., traffic signs). Additionally, "feature pyramid networks" (FPN) were used together with Faster-RCNN to compute convolutional neural network (CNN) feature maps at multiple-scales. This improves the recall and precision of the detections. FPN exploits the inherit multi-scale and pyramidal hierarchy of deep convolution networks to build high-level semantic feature maps at all scales with marginal computation cost (see [12]).

We explored Mapillary Traffic Sign (M-TS) [13], DFG Traffic Sign (DFGTS) [14], and German Traffic Sign Detection Benchmark (GTSDB) [15] datasets. Additionally, a detector trained on the DFGTS dataset was tested. While the detector trained on M-TS produced 6946 detections for roundabout signs, the detector trained on DFGTS detected only 4929 of those signs. Therefore, M-TS was chosen as the preferred training dataset.

To reduce implementation cost and time-to-integrate the trained model in overall experimental studies, we used the Object Detection API from Tensorflow [16]. The API provides several pre-implemented architectures with the ability to choose different deep convolution networks as a backbone (e.g., ResNet-152 and ResNet-101), along with pre-trained weights on ImageNet [17], COCO [18], PascalVOC [19] datasets.

The M-TS dataset contains images of different spatial sizes. Therefore, to facilitate batch training, we scaled all the training images to a fixed size. Image size is one of the hyperparameters for Faster-RCNN and had to be chosen carefully based on available GPU resource and needs. We resized all images at training to a fixed size of $1024 \times 1024$ pixels (i.e., width $\times$ height) along with maintaining the original aspect ratios of the image by zero-padding (when necessary). Furthermore, channel normalization with the computed mean RGB value on the entire training dataset was performed.

After careful evaluation, a variant of the residual neural network (ResNet) architecture with 101 layers, called ResNet-101 [20], was chosen as the backbone network. Using variants of ResNet, which are either shallower (e.g., ResNet-18, ResNet-50...etc.) or deeper than 101 layers (i.e., ResNet-152), did not bring any improvement in model performance.

The "region proposal network" (RPN) within Faster-RCNN computes a set of rectangular object proposals and objectness score (score belonging to foreground and background), by taking CNN maps as input. Default hyperparameter settings from [20] were chosen to produce *4K* and *2K* output which encodes the co-ordinates and score of objectness for each of the *K* proposals. More details on the hyperparameter settings used can be found in [11].

This finally led to a good recognition rate of the signs in the database. All traffic signs detected in the NWSIB street view images were added as geo-referenced entries to a database. They can be browsed in QGIS ([8]) (as shown in Figure 2) and filtered using a self-tailored QGIS plugin.

*2.4. Intersection of the Data*

Exactly 26 roundabouts from the 2011 dataset were re-identified for comparing crash and ADT numbers; see Table 1. Crash numbers were of the same order of magnitude, although [10] contains crashes with injured persons only, while [7] had been using all crashes, including crashes with material-only damage.

When looking at ADT values, it is apparent that very low ADT bike counts in the NWSIB dataset correspond to large deviations of bike counts between the NWSIB and the 2011 datasets. Ref. [7] state that they carried out manual traffic counts at the intersections for which no sufficiently up-to-date traffic demand data were available. This amounts to about 80 roundabouts where they carried out manual traffic counts. This gives rise to the assumption that their bike counts were more reliable than the NWSIB bike counts. Inductive loop bike detectors may especially significantly undercount bicycles when not calibrated properly. Therefore, the following two justifications were applied to the data:

- All roundabouts with qBike < 20 were removed from the NWSIB dataset, because it was suspected that their data were corrupt due to detector malfunction or calibration errors;
- For the NWSIB roundabouts that could be associated with the ones from the 2011 dataset and that exhibited qBike < 20, the ADT was replaced by the values from the 2011 dataset.

**Table 1.** Re-identified roundabouts in the dataset.

| fid | IsVeh.11 | IsVeh | IsBike.11 | IsBike | qBike.11 | qBike | qCar.11 | qCar |
|---:|---|---:|---|---|---:|---:|---:|---:|
| 4 | 2.67 | 1 | 0.7 | 1.0 | 709 | 119 | 10900 | 8324 |
| 25 | 0.67 | 0 | 0.0 | 0.0 | 347 | 21 | 6200 | 10,766 |
| 191 | 2 | 0 | 0.3 | 0.0 | 556 | 9 | 13,000 | 6194 |
| 297 | 12 | 3 | 2.3 | 0.5 | 1869 | 425 | 16,600 | 11,029 |
| 372 | 1.33 | 1 | 0.0 | 0.5 | 813 | 18 | 25,300 | 4310 |
| 449 | 5 | 0 | 1.5 | 0.0 | 438 | 259 | 13,500 | 9610 |
| 500 | 5.67 | 5 | 0.3 | 1.5 | 250 | 245 | 22,000 | 12,398 |
| 501 | 7.67 | 6 | 1.3 | 2.0 | 292 | 245 | 21,500 | 15,098 |
| 519 | 1.33 | 2 | 0.7 | 0.0 | 167 | 224 | 13,800 | 6808 |
| 539 | 3 | 2 | 0.3 | 0.5 | 1403 | 270 | 22,800 | 11,209 |
| 754 | 3.67 | 3 | 0.7 | 0.5 | 625 | 2 | 13,000 | 17,082 |
| 943 | 12.33 | 5 | 7.7 | 2.0 | 7072 | 2016 | 24,000 | 9246 |
| 945 | 8 | 9 | 2.0 | 3.5 | 3210 | 1198 | 6500 | 9583 |
| 946 | 1 | 1 | 1.0 | 0.0 | 2584 | 11 | 10,000 | 1082 |
| 995 | 7.33 | 4 | 0.4 | 2.0 | 3237 | 86 | 13,600 | 9121 |
| 999 | 5.33 | 3 | 1.0 | 1.0 | 2826 | 36 | 21,600 | 10,515 |
| 1021 | 4 | 5 | 0.0 | 0.5 | 1528 | 0 | 6600 | 11,776 |
| 1054 | 2.33 | 3 | 0.0 | 1.0 | 90 | 0 | 18,800 | 0 |
| 1067 | 1 | 1 | 0.0 | 0.0 | 345 | 748 | 12,200 | 20,076 |
| 1092 | 3,67 | 6 | 1.0 | 1.5 | 1014 | 18 | 17,000 | 11,264 |
| 1094 | 5 | 4 | 0.0 | 1.5 | 1389 | 1198 | 20,200 | 9583 |
| 1292 | 3 | 2 | 0.7 | 1.0 | 806 | 831 | 16,300 | 9891 |
| 1293 | 9.33 | 1 | 0.0 | 0.5 | 556 | 15 | 22,500 | 6993 |
| 1294 | 1 | 0 | 0.0 | 0.0 | 1612 | 341 | 18,000 | 1921 |
| 1296 | 2.5 | 2 | 0.0 | 1.5 | 855 | 5 | 7000 | 6670 |
| 1297 | 4.33 | 2 | 1.0 | 1.0 | 479 | 801 | 13,000 | 13,960 |

Compared to the sample drawn by [7], the present dataset has a bias towards small cities and rural areas. City areas are not well covered by the NWSIB dataset (See Figure 4). One relevant side effect might be that the mean ADTs of bikes and cars in our dataset are lower than the ones in [7]; see Table 2.

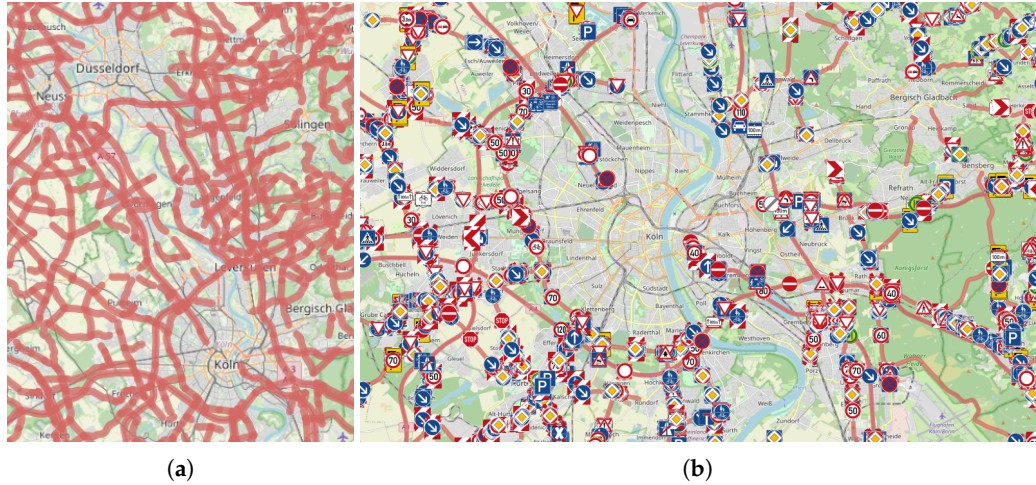

| (**a**) | (**b**) |

**Figure 4.** Metropolitan area coverage within the NWSIB Dataset: (**a**) Traffic volumes (marked in bright red) are not available in big cities such as Cologne and Dusseldorf; (**b**) Street view images (symbolized as detected road signs) are missing for the city of Cologne.

**Table 2.** ADT values for bicycles and cars in the two datasets for urban areas.

|  | A | B1 | B2 | B3 |
|---|---|---|---|---|
| Roundabouts NWSIB | 142 | 121 | 44 | 132 |
| Roundabouts of 2011 re-engineered | 29 | 35 | 11 | 25 |
| Roundabouts of 2011 Bondzio | 44 | 31 | 10 | 15 |
| $Q_{bike}$ of NWSIB | 175 | 222 | 209 | 202 |
| $Q_{bike}$ of 2011 re-engineered | 1098 | 1249 | 972 | 1123 |
| $Q_{bike}$ of 2011 Bondzio | 972 | 1512 | 750 | 1284 |
| $Q_{car} + Q_{bike}$ of NWSIB | 9304 | 10869 | 11,358 | 10,787 |
| $Q_{car}$ of 2011 re-engineered | 17,234 | 17,378 | 14,945 | 14,931 |
| $Q_{car} + Q_{bike}$ of Bondzio | 13,913 | 17,753 | 14,935 | 16,423 |

## 3. Results

For each roundabout, the following information has been gathered:

- Its position;
- The crashes, of which the information about the involved traffic objects has been used. The database contained the crashes with injured persons for the years 2019 and 2020, and is publicly accessible [10]. They are coded as `IsVeh` and `IsBike`;
- The ADT-values, coded as `qCar` and `qBike`, or in the equations, as $Q_{car}$ and $Q_{bike}$, respectively;
- Additional information about each roundabout, such as:

  - The location (within city-limits/ urban or outside/ rural), the study of [7] was only on roundabouts within city-limits;
  - Whether the bike crossing was colored in red or not;
  - The existence of a zebra crossing;
  - A warning sign to motorized traffic "Careful, bikes" (German traffic sign "Z 138");
  - Information about if the bike lane was two way (two directions);
  - Very rarely: if the geometry or the organization had been changed; there was an attempt to extract this information manually from Google Earth's timeline.

### 3.1. Comparison with the Results of the 2011 Study

The data were analysed in different ways. Of course, one goal was to reproduce (or to falsify) the results of [7], and by utilizing the additional statistical power, to obtain additional insights.

Comparing our results with those of [7] is not completely straightforward, for a number of reasons that are difficult to reconcile.

- There was no access to the original data, but in fact only to the parts that have been published in the report. e.g., their detailed classification data are lacking;
- Bondzio at al. have used the total number of crashes, while the German crash database lists only severe crashes, i.e., those where crash participants have been injured or even killed.

In Figure 5, the findings of this study are compared against the results of [7]. Only the 439 roundabouts in urban areas have been used from the sample of 1300 roundabouts, of course, and crash rates were computed for the comparison as advised in [7] (there are other results in this study):

$$r = \frac{N_{car} + N_{bike}}{Q_{car} + Q_{bike}}. \tag{1}$$

While the crash-rates are similar (which is unexpected, because they used all crashes, but this study used only the severe crashes), a number of differences could be noted between this study and that of [7]. In addition, this study analyzes the bike crashes.

The most dangerous roundabout type (for cars) is B1 while it was B2 before. However, the difference between B1 and B2 may not be statistically significant in [7], while here it is at the 5% level: a Kolmogorov–Smirnov (KS-) test yields $D = 0.25$, $p = 0.035$ between B1 and B2. Ref. [7] found type A to be the safest roundabout, while this study has B3 as the safest; however, further KS-tests indicate that the types A, B2, and B3 are in one safety class, while B1 is in its own class and statistically less safe than all the others. It will be shown below that this result has to be put into perspective given additional analytical results.

There is one issue that could not be resolved: the standard deviation (of the rates) is much larger than that of [7]. Interestingly, Bondzio et al. used the confidence intervals anyway, so Figure 5 uses this to compare our results to those of [7]. The larger standard deviation is probably due to the fact that the method of assigning the ADT-values produces a larger scatter in the replication study than in the original one.

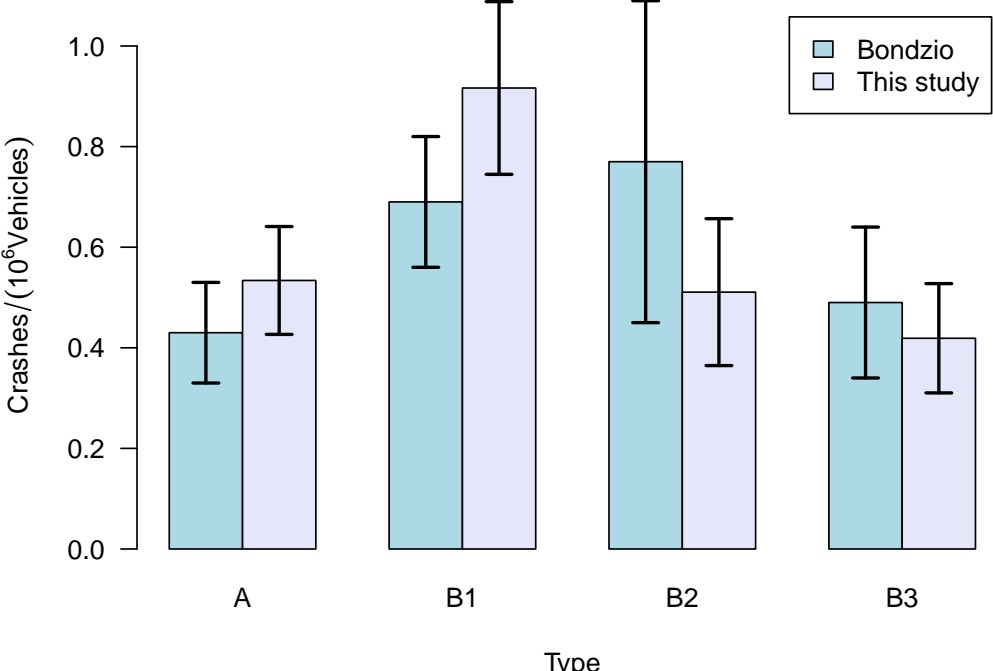

**Figure 5.** Comparison of the results of this study with the results of [7]. The rates have been scaled so that the mean values fit; the scaling factor was 1.61.

Note, too, that a KS-test of the rates reveals that there is a highly significant difference ($D = 0.155$, $p = 0.001$) between a roundabout within city-limits or outside. However, this result is not robust; it depends on how the rates are computed, in this case as in Equation (1). When using cars-only and bikes-only rates, the difference between urban and rural becomes statistically smaller.

The crash rates of this study are similar to those reported by [7] for the types of roundabouts A and B1. The larger deviations for B2 and B3 types may stem from misclassifications: it is sometimes difficult to differentiate between type B2 and type A roundabouts. Every time the signs "Z 240" or "Z 241" and a zebra crossing could be seen, it was concluded that bicycles are supposed to go on the sidewalk and the roundabout was classified as B1. Ref. [7] might have been classifying those roundabouts as A (cp. Section 2.1).

Note that [4] found fairly different rates for the cars and bikes, respectively, so the rates may not be robust between different countries or different studies.

It should be noted that the rates, but not necessarily the ranking of the roundabouts, are very different when car and bike safety is compared, see Figure 6 for this. This result is also in line with [4,5,7].

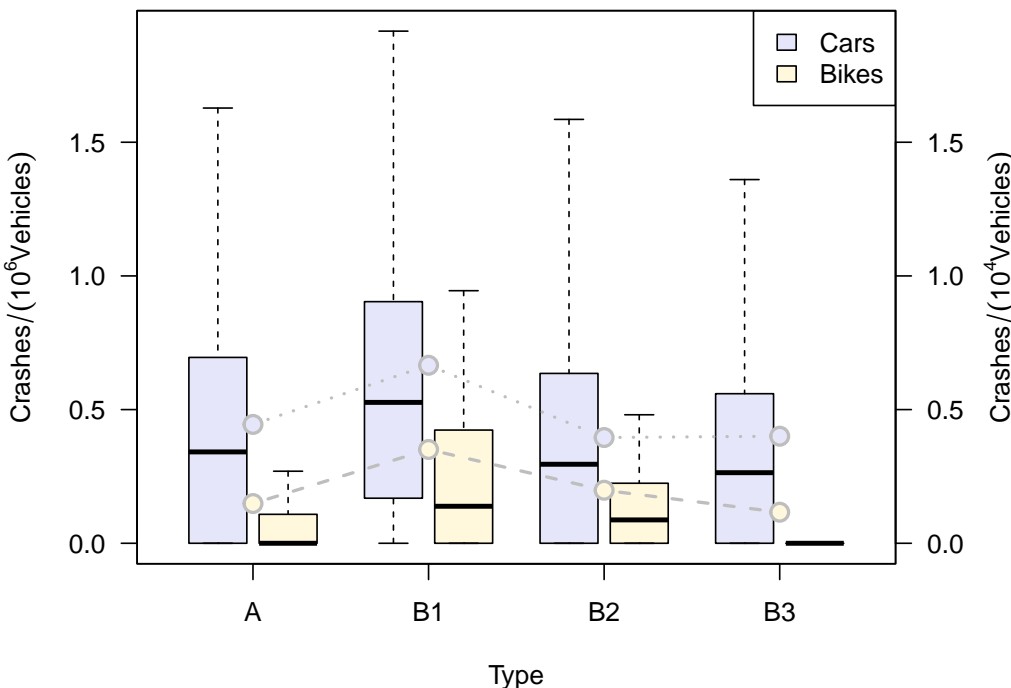

**Figure 6.** Comparison of the crash-rates of cars and bikes. Note the factor 50 in the scaling of the two y-axes. The horizontal dashed lines connect the mean values.

### 3.2. Poisson versus Negative Binomial

In the following, models will be used to gain additional insight. All analyses have been performed with the help of [21]. As usual, it is assumed that the mean value of the number of crashes depends on the ADTs and other factors such as (see [4,22–26]):

$$\mu = \beta_0 Q_{car}^{\beta_1} Q_{bike}^{\beta_2} \exp\left( \sum_{i=3}^{K} \beta_i x_i \right), \tag{2}$$

where $\mu$ is the mean value of either a Poisson or a negative binomial distribution, and the $x_i$ are the factors described earlier in this paper (Type, ford.colored, zebra, care.bike, both.dir, location). The $Q$'s are the ADT-values; of course it would be much better to have the actual flow-rate at the time of each crash, which was not available.

Sometimes, it is useful to analyse the corresponding rates, i.e., $r = \mu/Q$ (see [27]):

$$r_{car} = \frac{\mu}{Q_{car}} = \beta_0 Q_{car}^{\beta_1 - 1} Q_{bike}^{\beta_2} \exp\left( \sum_{i=3}^{K} \beta_i x_i \right), \tag{3}$$

and similar for the bike-rate $r_{bike}$. Note that these rate equations are slightly more involved when fitted against the data. Nevertheless, the `glm()` algorithms in R provide an excellent toolbox to deal with rates as well, namely by using $log(Q_x)$ as an offset.

Before going into the details of how to identify the correct model family, in a first step the underlying distribution is investigated. This is done by estimating a model both with a Poisson (P) and a negative binomial (NB) distribution and comparing the model-quality.

The differences between a Poisson and a negative binomial distribution are not large when viewed in terms of the resulting coefficients. The coefficients of the NB have a slightly wider distribution, see Figure 7. However, the fit also results in a value of the size parameter (it has different names in different settings, `glm.nb()` in R calls it $\theta$) of the NB distribution of about three ($\theta = 3.2 \pm 0.6$, for the bike crashes: $\theta = 1.7 \pm 0.4$), where $\theta$ is the denominator in the relationship between mean value $\mu$ and the variance $\sigma^2$ of the NB distribution, $\sigma^2 = \mu + \frac{\mu^2}{\theta}$.

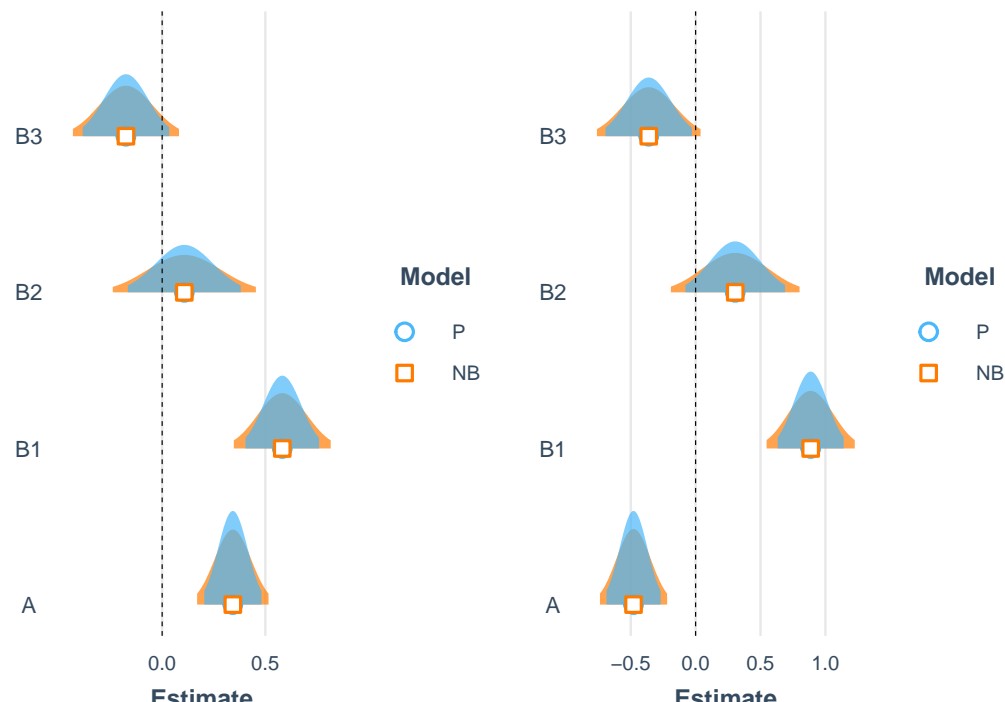

**Figure 7.** Uncertainty of Parameter values of the GLM (Poisson versus Negative Binomial Fit). **Left** for the cars, **right** for the bikes.

For $\theta \to \infty$ this yields a Poisson distribution, so the $\theta$ found here is considerably different from infinity and indicates in fact that the NB is a better fit than the Poisson distribution. This is confirmed by ANOVA tests and by computing the AICs for the different models: the residual deviance drops by a large amount when switching from P to NB, both for the bikes as well as for the cars. The same holds for the AICs, where a drop by 128 (cars) and 86 (bikes), respectively, has been observed. This is the result for all the roundabouts, it does not change much when analysing only the urban roundabouts. Therefore, the rest of the paper works with an NB distribution only.

It might be noted in passing that this is not in line with the results of [4], who found under-dispersion for the crashes at roundabouts.

Note, too, the different dangerousness of roundabouts of Type A for cars and bikes, respectively. While the other types are similar between bikes and cars, Type A is difficult: it seems to improve safety for cars, but to decrease it for bikes.

### 3.3. Exposure

Another interesting point (and a bit puzzling as well) is the relationship between the number of crashes and the exposure measured by the ADT-values. In a first step, the two relationships between $N_x$ and $Q_x$ are estimated, where $x$ is cars or bikes, respectively.

They are discerned by location, yielding the results displayed in Figure 8. It indicates that the number of crashes $N$ grows sublinearly with the number of objects $Q$, both for the cars as well as for the bikes. The exponents $\beta_i$ for the two different locations in $N = \alpha Q^{\beta_i}$ are $\beta_i = 0.45, 0.56$, each with an error of 0.1 (rural first). The values for the bikes are similar $\beta_i = 0.77, 0.37$ with an error of 0.09. All exponents are highly significant, the worst $p$-value is $3 \times 10^{-6}$ for the rural car $\beta$.

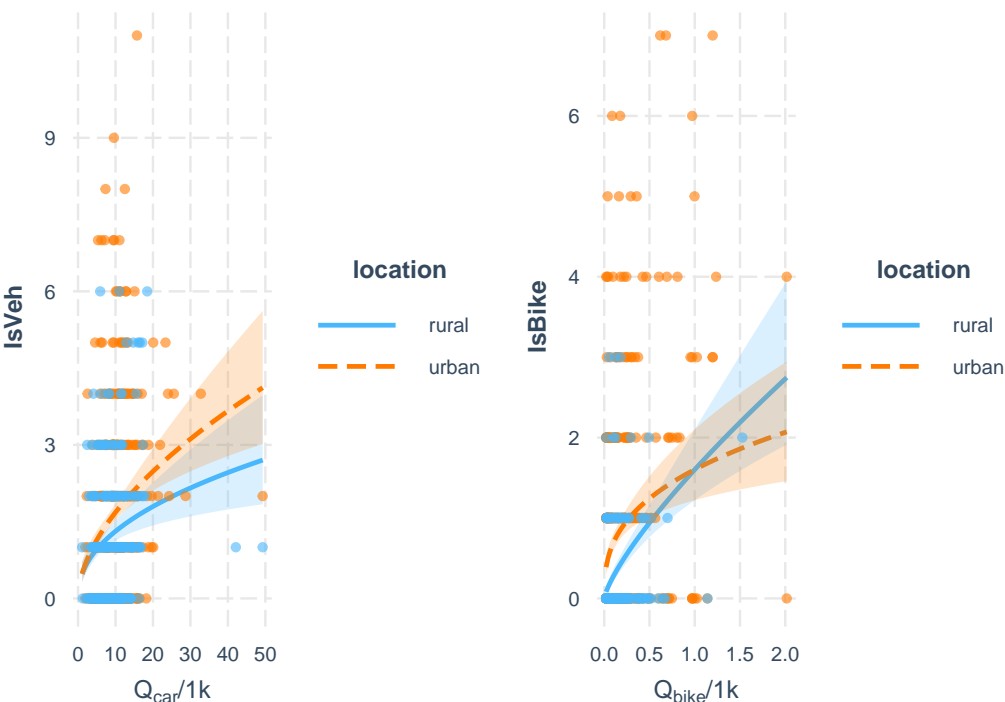

**Figure 8.** Number of crashes fitted versus the ADT values, classified by location.

This result is puzzling, since it indicates that the corresponding rates decrease with increasing demand: while not entirely impossible, and sometimes interpreted as a safety-in-numbers effect, it is not easy to argue why this should be the case (especially since $r \to \infty$ for $Q \to 0$).

One may notice that roundabouts with a large number of crashes are not found in this dataset, instead there are a few roundabouts with a large exposition and a small number of crashes. This might be due to an error in the way the data have been fused, or it might be a problem with the estimation of the ADT-values itself, which is beyond the scope of this study. Even by restricting the fit to smaller values of $Q_{\text{car}} < 20,000$ does not increase the exponent to a value larger than one, so this result is robust and does not hinge on a few outliers.

These results are at least not in contradiction with the results of [4]. They found exponents for the cars and bikes in the range of $\beta_{\text{car}} = 0.89 \ldots 1.16$ and $\beta_{\text{bike}} = 0.27$. Interestingly, some of the raw data of [7] are printed in the appendix of their report and allow an analysis of $N = \beta_0 Q_{\text{car}}^{\beta_1}$, but only for the car-data. In this case, a clear super-linear growth results, where $\beta_1 = 1.54$. Note that [4,7] have used all crashes in their analysis, while here only the severe crashes have been used.

It might be noted in passing that by fitting the rates instead of the number of crashes, the very same results are obtained.

There is a final remark regarding the exponents of the full model, which claims that the number of crashes depends on the product of $Q_{\text{car}}^{\beta_1} Q_{\text{bike}}^{\beta_2}$. Here, one would expect at least $\beta_1 + \beta_2 > 1$, however, this is not the case. Instead, the values of the exponents are lower:

$$N_{\text{car}} \propto Q_{\text{car}}^{0.43} Q_{\text{bike}}^{0.20}$$

and

$$N_{\text{bike}} \propto Q_{\text{car}}^{0.37} Q_{\text{bike}}^{0.44}.$$

Note that the values of the exponents $\beta_1, \beta_2$ are not very robust; small changes in the models cause changes in the exponents. However, the result that their sum is smaller than

one is robust, and all coefficients are statically significant, with the worst *p*-value of 0.02 for $Q_{car}$ in the bike model.

### 3.4. The 2011 Model

The list of 100 roundabouts in [7] was reviewed; the type of roundabout, the presence of a zebra crossing and a red painted ford in 2011 were added as additional information. Table 3 shows the model parameters. Obviously there was a discrepancy in the typing compared to the original paper (see Table 2).

**Table 3.** GLM model fit for the re-engineered dataset with 100 roundabouts used by [7]. Parameter values that are statistically significant with 5% probability of error are printed in bold letters.

|  | Cars | | | Bikes | | |
|---|---|---|---|---|---|---|
|  | **Estimate** | **Std. Error** | **Pr (>\|z\|)** | **Estimate** | **Std. Error** | **Pr (>\|z\|)** |
| (Intercept) | −0.803 | 1.777 | 0.651 | **−12.705** | **4.302** | **0.003** |
| I(log(qCar.11)) | 0.177 | 0.178 | 0.320 | 0.789 | 0.433 | 0.068 |
| I(log(qBike.11)) | 0.061 | 0.078 | 0.434 | **0.633** | **0.158** | **0.000** |
| TypB1 | **0.430** | **0.203** | **0.034** | 0.037 | 0.352 | 0.916 |
| TypB2 | 0.174 | 0.257 | 0.500 | −0.718 | 0.686 | 0.296 |
| TypB3 | −0.304 | 0.247 | 0.217 | −0.652 | 0.661 | 0.323 |
| Zebrastr.TRUE | **−0.536** | **0.226** | **0.018** | 0.532 | 0.531 | 0.317 |
| Furt.eingefärbtTRUE | **0.803** | **0.224** | **0.000** | 0.415 | 0.390 | 0.288 |

The following was observed for the fit displayed in Table 3:

- The overdispersion was low for cars ($\theta = 6.99$) and bikes ($\theta = 5.99$);
- The intercept in the model for car crashes is not significantly different from zero, while type B1, zebra crossing and ford painted are significant. The effect of the zebra crossing variable is negative;
- Surprisingly, in the model for car crashes, the qBike.11 and qCar.11 variables are not significant;
- In the model fit for bike crashes, only the qBike.11 is significant.

### 3.5. The Full Model

In a final step, a model that includes all the influences is estimated, both for the cars as well as for the bike crashes.

The results are displayed in Table 4. The scale parameters were: for car crashes $\theta = 4.1$, and for bike crashes $\theta = 2.0$.

The results show that not all of the coefficients are statistically significant. Most notable is (for the car crash numbers) the existence of a zebra crossing (it increases the crash-numbers), a colored ford (increases crash-numbers, too), and the exposition by $Q_{car}$. While the exponent is similar to the results above, in this case the exponent of the bike exposure is statistically weak; it barely hit 5% significance. From the perspective of the cars, the classification of the types introduced by Bondzio is only interesting as it discerns the Type A from the rest.

For the bikes, the same coefficients are significant; this time, however, the typology is more meaningful since at least the value for B1 is significant.

The most surprising result in our view is that both for the bike-crashes as well as for the car-crashes, the existence of a zebra-crossing and of a colored ford increase the number of crashes. In fact, additional t-tests confirm these results. We will come back to this issue later on.

**Table 4.** The full model. Parameter estimates for car crashes (left), and bike crashes (right). Parameter values that are statistically significant with 5% probability of error are printed in bold letters.

|  | **Cars** | | | **Bikes** | | |
| --- | --- | --- | --- | --- | --- | --- |
|  | **Estimate** | **Std. Error** | **Pr (>\|z\|)** | **Estimate** | **Std. Error** | **Pr (>\|z\|)** |
| (Intercept) | **−3.981** | **0.802** | **0.000** | **−4.929** | **1.322** | **0.000** |
| TypeB1 | 0.220 | 0.127 | 0.083 | **0.479** | **0.188** | **0.011** |
| TypeB2 | 0.053 | 0.162 | 0.743 | 0.283 | 0.237 | 0.232 |
| TypeB3 | 0.148 | 0.130 | 0.255 | 0.043 | 0.215 | 0.841 |
| locationurban | −0.102 | 0.099 | 0.301 | **0.479** | **0.181** | **0.008** |
| zebraTRUE | **0.491** | **0.128** | **0.000** | **0.466** | **0.193** | **0.016** |
| ford.coloredTRUE | **0.336** | **0.143** | **0.019** | **0.545** | **0.198** | **0.006** |
| care.bikeTRUE | 0.155 | 0.285 | 0.587 | 0.014 | 0.459 | 0.975 |
| both.dirTRUE | −0.096 | 0.116 | 0.405 | −0.243 | 0.195 | 0.213 |
| pedXTRUE | 0.070 | 0.109 | 0.520 | 0.160 | 0.160 | 0.316 |
| kph30an | 0.023 | 0.134 | 0.863 | −0.042 | 0.202 | 0.836 |
| kph30in | −1.294 | 0.779 | 0.097 | -0.615 | 0.859 | 0.474 |
| log(qCar) | **0.373** | **0.091** | **0.000** | 0.212 | 0.151 | 0.160 |
| log(qBike) | **0.144** | **0.039** | **0.000** | **0.340** | **0.063** | **0.000** |

## 4. Cross-Validation

This section discusses the various models from the perspective of cross-validation. Its usage in road accident analysis is so far limited; it was demonstrated, however, that cross-validation techniques can be useful for model selection for predicting the crash severity of vulnerable road users [28]. It is also known from the literature that a model trained on a training dataset may perform poorly when predicting the value of the dependent variable on a disjointed test dataset (see [29]).

Considering Table 4, it might be expected that a model that accounts for the variables that are statistically significant and that was trained on some training dataset should do well in predicting annual crash numbers on a test dataset. The ability to do this was tested in extensive cross-validation runs in this study.

The motivation for cross-validation experiments was the following: It can be assumed that the model that best predicts accident rates in places it has never seen before should only include factors that really have an impact on accident rates. Those factors are the ones that traffic planners should consider when designing safe infrastructure.

### 4.1. K-Fold Cross Validation

About 2/3 of the whole dataset was used for training, thus fitting the GLM model, and the remaining 1/3 for testing. Testing was carried out by letting the model predict the annual crash numbers for each single roundabout in the test dataset and comparing the estimate with the actual crash number. As a metric, the mean square error (MSE) was used. For assessing the prediction performance, the MSE for 900 random test/training partition runs and the moments and quantiles of the MSE were evaluated.

Figure 9 shows the results of cross validation for the following GLMs:

- Q: The number of bike crashes depends on the average daily traffic of bikes and motorized road users;
- Q.AB123: Just as Q, but in addition considers the type of roundabout: A, B1, B2, or B3;
- Q.AB123.Z.F: Just like Q.AB123, but in addition considers if there is a zebra crossing and a ford colored;
- Q.AB23_B1.Z.F: Just as Q.AB123.Z.F, but type differentiation is only between B1 and {A;B2;B3}. The motivation for creating this model is that, in Table 4, only the difference between roundabout type B1 and all others is statistically significant;
- Q.AB23_B1.Z.F: Just as Q.AB123.Z.F, but type differentiation is only between the groups {B1;B2} and {A;B3}. The motivation for creating this model is that B1 and B2 roundabouts grant right of way to bicycle riders. We suspect that this right of way

regulation has an effect on crash numbers. In addition, as can be seen in Table 5, the level of significance of factor "TypeB12" of this model is higher ($\alpha < 1.5\%$) than the significance level of factor "TypeB1" of Model Q.AB123.Z.F ($\alpha < 3.5\%$). See Table 4;

- Q.Z.F.: like Q, considers if there is a zebra crossing and a colored ford but does not care if it is of Type A, B1, B2, or B3.

**Table 5.** The model with the best prediction performance (Q.AB3_B12.Z.F) for bike crashes

|  | Estimate | Std. Error | z Value | Pr(>\|z\|) |
|---|---|---|---|---|
| (Intercept) | −3.122 | 0.315 | −9.903 | 0.000 |
| log(qBike) | 0.363 | 0.060 | 6.079 | 0.000 |
| TypeB12 | 0.390 | 0.156 | 2.505 | 0.012 |
| zebraTRUE | 0.540 | 0.140 | 3.846 | 0.000 |
| ford.coloredTRUE | 0.569 | 0.185 | 3.080 | 0.002 |
| locationurban | 0.501 | 0.174 | 2.880 | 0.004 |

With the help of Table 6 and Figures 9 and 10 the following is observed:

- The prediction accuracy of a model that introduces the type of roundabout is higher than of a model that only accounts for exposure of the average daily traffic volume of bicycle riders and motorized road users. This provides evidence that there is an effect beyond exposure. Note that roundabouts of type B1, on average, also have a higher average daily traffic volume of bikes. This gave rise to the suspicion that the different levels of crash rates at the roundabout types A, B1, B2, and B3 may simply come from a selection bias. Obviously, this is not true;
- The prediction accuracy of model Q.Z.F is higher than for Q.AB123.Z.F. This observation is notable and gives rise to the assumption that the differentiation between A, B1, B2, and B3 roundabouts is less meaningful than expected. Instead, the predictive power of Q.AB123 seems to stem from correlations between the types A, B1, and B2 and the presence of a zebra crossing and a colored ford. The roundabout types do not add meaningful information to the GLM. Instead, they seem to introduce overfitting;
- Although the difference of type B1 and A (Intercept) is statistically significant, model Q.AB23_B1.Z.F is less accurate than Q.Z.F. This demonstrates one of the challenges of proper model selection. Only when differentiating between roundabouts of type {B1;B2}, which give priority to cycles and {A;B2}, which do not, improves prediction performance. This is true, although roundabouts of type B2 do not statistically significantly differ from roundabouts of Type A (the Intercept).

While [29] reported on such types of effects, there were still doubts that these findings were correct.

**Table 6.** K-Fold cross validation prediction errors for crashes of bicycles and of cars.

|  | Q | Q.AB123 | Q.AB123.Z.F | Q.AB23_B1.Z.F | Q.AB3_B12.Z.F | Q.Z.F |
|---|---|---|---|---|---|---|
| MSE Bike | 1.088 | 0.993 | 0.958 | 0.954 | 0.948 | 0.944 |
| SD Bike | 0.177 | 0.160 | 0.142 | 0.149 | 0.138 | 0.139 |
| AIC Bike | 1359.442 | 1321.098 | 1310.053 | 1307.608 | 1306.703 | 1310.936 |
| MSE Car | 2.433 | 1.003 | 0.965 | 0.960 | 0.954 | 0.950 |
| SD Car | 0.337 | 0.164 | 0.148 | 0.147 | 0.154 | 0.149 |
| AIC Car | 2190.346 | 2159.660 | 2143.581 | 2140.780 | 2142.411 | 2141.721 |

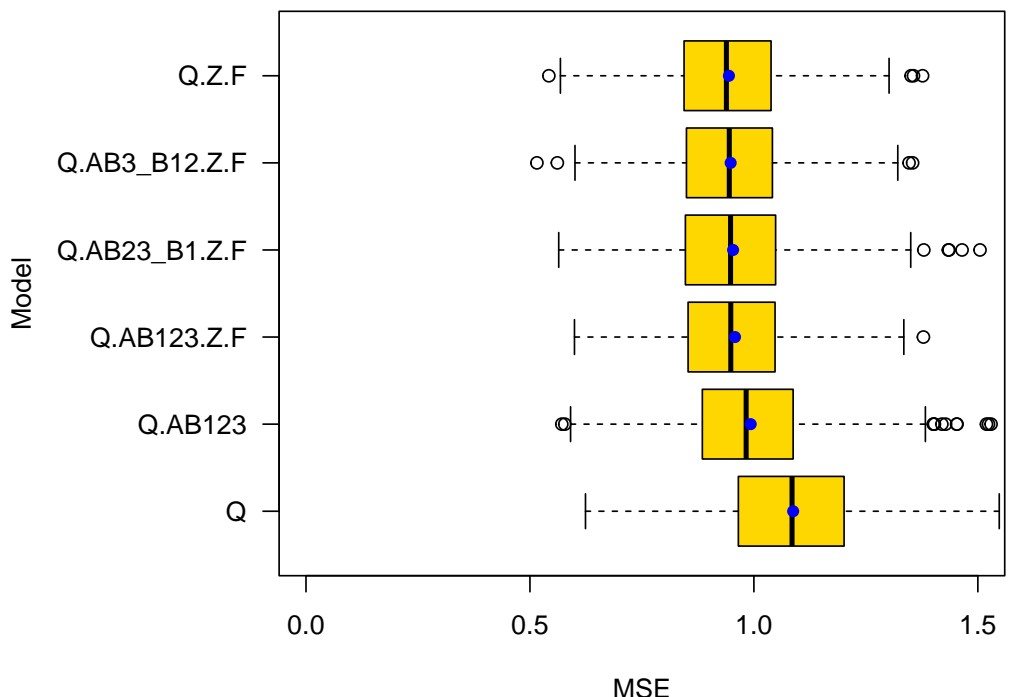

**Figure 9.** K-Fold cross validation results for bike crashes. The blue dots denote mean values. The bold black lines mark the 50% percentile, thin black lines the 5% and 95% percentiles respectively and the black circles denote outliers.

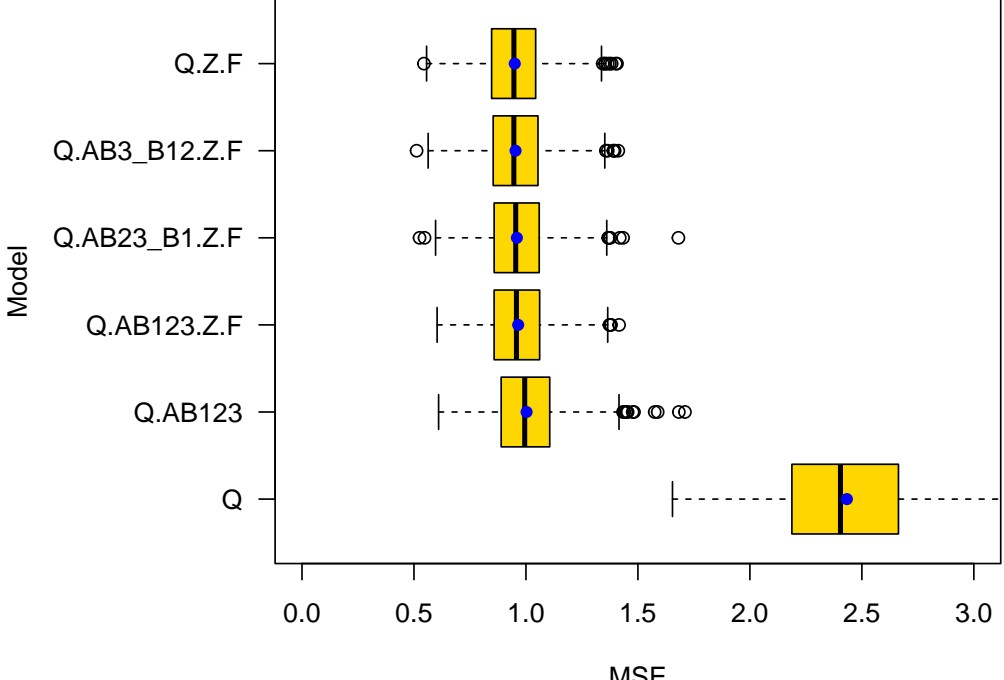

**Figure 10.** K-Fold cross validation results for car crashes. The blue dots denote mean values. Again, the bold black lines mark the 50% percentile, thin black lines the 5% and 95% percentiles respectively and the black circles denote outliers.

*4.2. Cross Validation on Ensembles*

Mean square error calculations on data with high dispersion can produce misleading results because square error puts more weight on predictions that show stronger deviation

from the ground truth. To circumvent this potential problem, the procedure was modified so that the model would estimate mean accident rates for ensembles.

Table 7 illustrates ensemble validation. It can be seen in the first row, that for all 94 roundabouts of type A in urban area that have a zebra crossing, the ensemble mean ADTs are 199 bicycles and 9439 cars. Given those numbers, the GLM "Q.Z.F" predicts an average number of 1.03 bike crashes during the observation period of two years. Actually, there were observed 0.78 crashes on average during the same period of time. When looking at the predicted and actual numbers it becomes obvious that the model predicts the expected crash numbers well.

**Table 7.** Prediction of crash numbers for ensembles in the complete dataset. The Model is Q.Z.F

| Type | Zebra | Ford | Location | # | Q_car | Q_bike | Pred | Real | CR_bike |
|------|-------|------|----------|-----|--------|--------|------|------|---------|
| A | TRUE | FALSE | urban | 94 | 9439 | 199 | 1.03 | 0.78 | $2 \times 10^{-05}$ |
| A | FALSE | FALSE | urban | 48 | 8522 | 128 | 0.46 | 0.31 | $9 \times 10^{-06}$ |
| A | FALSE | FALSE | rural | 28 | 7881 | 119 | 0.25 | 0.18 | $7 \times 10^{-06}$ |
| B1 | TRUE | FALSE | urban | 61 | 10,641 | 165 | 0.96 | 1.15 | $3.5 \times 10^{-05}$ |
| B1 | FALSE | TRUE | urban | 10 | 9093 | 191 | 1.17 | 0.6 | $1.3 \times 10^{-05}$ |
| B1 | TRUE | TRUE | urban | 38 | 11,368 | 347 | 2.75 | 2.5 | $4.4 \times 10^{-05}$ |
| B1 | FALSE | FALSE | urban | 12 | 9693 | 140 | 0.48 | 0.92 | $3.5 \times 10^{-05}$ |
| B2 | TRUE | FALSE | urban | 22 | 11,209 | 216 | 1.06 | 0.77 | $1.9 \times 10^{-05}$ |
| B2 | FALSE | FALSE | urban | 20 | 10,125 | 180 | 0.52 | 0.75 | $2.3 \times 10^{-05}$ |
| B3 | FALSE | FALSE | urban | 132 | 10,585 | 202 | 0.54 | 0.43 | $1.5 \times 10^{-05}$ |
| B3 | FALSE | FALSE | rural | 187 | 9408 | 142 | 0.27 | 0.25 | $9 \times 10^{-06}$ |

In the ensemble cross validation experiments the GLM was trained on a training dataset that consists of 2/3 of all roundabouts randomly selected. The prediction procedure was performed for the test dataset in the same way as shown in the Table 7. The test dataset consists of the rest of roundabouts. To ensure effective averaging, a minimum of three samples per ensemble was required (column # in Table 7). Each time the ensemble has less than three samples, the predicted number of accidents is not scored.

A takeaway from Table 7 is that looking at crash rates of ensembles, as [7] did, produces misleading results. For the ensemble of Type B2 with a zebra crossing the average bicycle crash rate is lower ($CR_{bike} = 1.9 \times 10^{-5}$ crashes per bicycle) than for the ensemble without ($CR_{bike} = 2.3 \times 10^{-5}$). Ref. [7] might have concluded that a zebra crossing makes the roundabout safer for cyclists. This is not true, even with the dataset of 100 roundabouts that they used in 2011. The GLM states that the exposure-adjusted risk of a bicycle crash is $e^{0.47}$ times, thus 59% higher in the presence of a zebra crossing (see Table 5). This value is well in line with the effect size that a GLM fit would report for the 2011 dataset containing 100 roundabouts: $e^{0.53}$ or 69% higher (see Table 3). In the dataset from 2011, however, the effect is not significant.

A similar problem is apparent for B1 roundabouts without a zebra crossing that do and do not have a colored ford. Those roundabouts with a colored ford show the lowest crash rate of all B1 sub-ensembles ($CR_{bike} = 1.3 \times 10^{-5}$). Again, painting a bicycle ford does not reduce crash rates, as can be seen in the present dataset as well as in the 100 roundabouts dataset from 2011. The exposure-adjusted risk of a bicycle crash is $e^{0.56}$ times, thus 75% higher in the presence of a colored ford. This value, again, is well in line with the effect size that a GLM fit would report on the 2011 dataset of 100 roundabouts: $e^{0.42}$ or 52% higher (see Table 3), while also not significant.

There are two possible reasons for that extent of error with empirical rates. First, there are locations with zero crashes within the observation period: for a place with 1000 bicycles ADT, zero crashes have another weight, than for a place with 10 bicycles ADT. GLMs account for this difference. Empirical crash rates do not. Second, crash numbers do not linearly rise with ADT. The mean ADT of bikes is nearly twice as high for B1 locations with

a zebra crossing and a colored ford compared to locations without. Comparing crash rates is synonymous with assuming that $\mu \propto Q_{bike}^{1.0}$ while it is actually $\mu \propto Q_{bike}^{0.37}$ (see Section 3.5).

A review of k-fold cross-validation and ensemble cross-validation yields the following findings (see Figures 11 and 12, Table 8) :

- Ensemble cross validation achieved more precise predictions. The MSE of the best performing model Q.AB3_B12.Z. is three (for cars) to five (for bikes) times lower in ensemble cross-validation than in k-fold cross validation;
- The prediction performance differences are bigger in ensemble cross validation. The k-fold cross validation prediction performance of the two best performing models Q.AB3_B12.Z.F and Q.Z.F differs by 0.4% in their mean squared prediction error for both car and bike crashes. It differs much more in ensemble cross validation: 2.1% for car and 6.8% for bike crashes. These differences are notable. They are in line with expectations, because the "TypeB1" variable is significant for bikes while closely missing the 5 significance level for car crashes;
- The best performing model is different in ensemble (Q.AB3_B12.Z.F) and k-fold (Q.Z.F) cross validation. Q.AB3_B12.Z.F cares if vulnerable road users have right of way priority, Q.Z.F does not. However, since the prediction performance differs by as much as 6.8% for bike crashes in ensemble cross validation compared with only 0.4% in k-fold cross validation, there is evidence that Q.AB3_B12.Z.F is the best performing model for predicting bike as well as car crashes. The parameter values of model Q.AB3_B12.Z.F can be seen in Table 5. Clearly, right-of-way is important for the road safety of vulnerable road users;
- While in ensemble cross validation the MSE is much lower than in k-fold cross validation, the standard deviation of the squared errors is higher, and even higher than its mean;
- The 50% and 75% quantiles of the squared errors in ensemble validation are very low compared with their mean. In contrast, k-fold cross validation yields quite similar median and mean values. Traffic engineers using ensembles for predicting crash numbers might be very accurate in most (e.g. 75%) of their attempts to do so;
- For bike crashes, the model with the lowest AIC (Q.AB3_B12.Z.F) performs best in cross validation. However, this is not true for car crashes. For car crashes, Q.AB23_B1.Z.F has the lowest AIC but Q.AB3_B12.Z.F achieves the most precise predictions;
- The models that rely on statistically significant variables only (Q.AB23_B1.Z.F for bikes and Q.Z.F for cars) do not yield the best generalization capability.

**Table 8.** Cross validation prediction errors for crashes of bicycles and cars. Ensemble approach.

|          | Q | Q.AB123 | Q.AB123.Z.F | Q.AB23_B1.Z.F | Q.AB3_B12.Z.F | Q.Z.F |
|----------|------|---------|-------------|---------------|---------------|----------|
| MSE Bike | 0.317 | 0.220 | 0.193 | 0.249 | 0.175 | 0.187 |
| SD Bike | 0.703 | 0.453 | 0.414 | 0.505 | 0.386 | 0.402 |
| AIC Bike | 1359.442 | 1321.098 | 1310.053 | 1307.608 | 1306.703 | 1310.936 |
| MSE Car | 0.598 | 0.568 | 0.356 | 0.493 | 0.328 | 0.335 |
| SD Car | 1.080 | 0.927 | 0.646 | 0.799 | 0.589 | 0.603 |
| AIC Car | 2190.346 | 2159.660 | 2143.581 | 2140.780 | 2142.411 | 2141.721 |

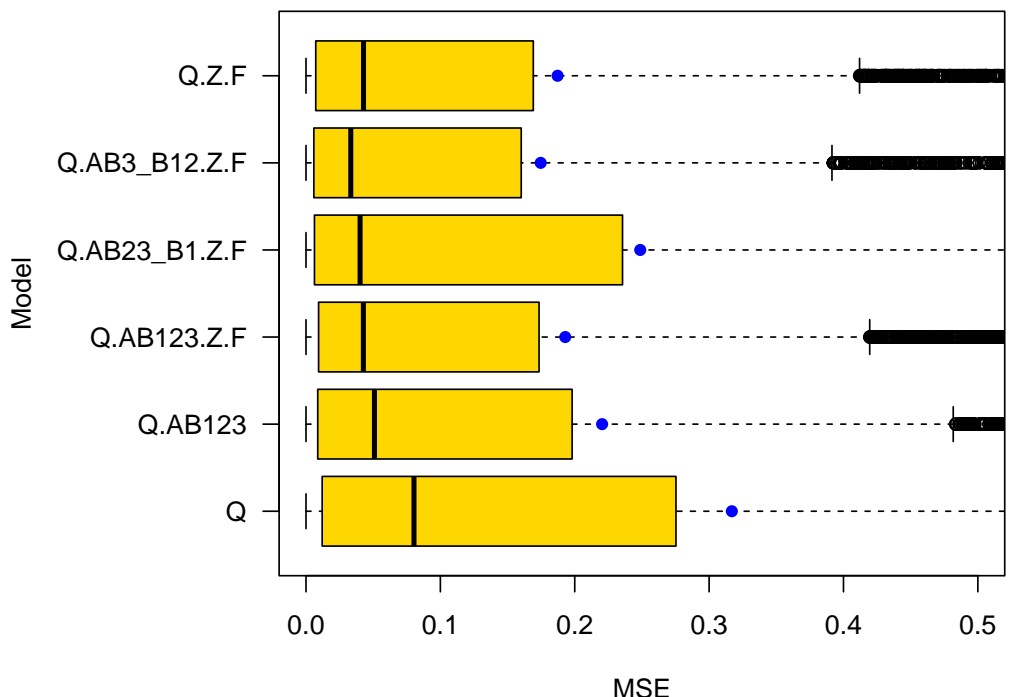

**Figure 11.** Cross validation results for number of bike crashes. Ensemble approach. The bold black lines mark the 50% percentile, thin black lines the 5% and 95% percentiles respectively and the black circles denote outliers.

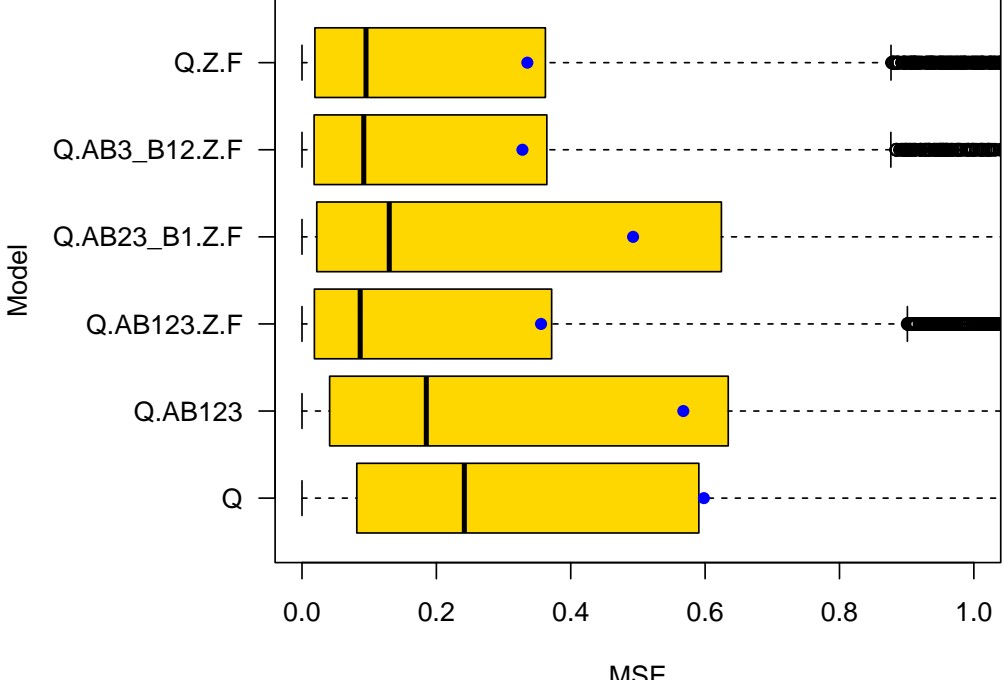

**Figure 12.** Cross validation results for number of car crashes. Ensemble approach. The bold black lines mark the 50% percentile, thin black lines the 5% and 95% percentiles respectively and the black circles denote outliers.

## 5. Conclusions

This analysis has shown that an approach based on machine learning for gathering data from difficult or expensive to access databases can help to gather large amounts of data that improve the statistical power of traffic safety investigations. We were able to

reproduce some, but not all, of the results from other studies, and have found useful and interesting differences to other studies.

A number of results were obtained that we believe require more thorough investigation. For instance, one very clear result of this study is that roundabouts where the bicycle ford is painted in red are more dangerous than roundabouts where this is not the case. It is not very likely that this is in fact a causal relationship, instead the causation might be the other way round: a dangerous roundabout has its bike-fords painted red, and this analysis just sees the result of such an intervention. However, without going into the history and investigating in fact whether this is the case or not needs to be turned out. Of course, if in fact red bike-crossings are more dangerous, then appropriate actions should be taken.

The additional information that can be gained from a cross-validation approach is also interesting. The full model with all the probably influencing factors in Table 4 had many statistically weak parameters. The approach based on cross-validation yields a much more concise model with a reduced parameter set, where all parameters are significant, see Table 5 for a direct comparison.

This, finally, points to a better recording of what has happened to the road infrastructure, and to make open those databases. Then it might be easier to learn and to find better solutions that increase traffic safety for all road-users, and especially for the vulnerable ones.

For practitioners, the most important finding may be that differentiating roundabout types has not been shown to add value. The different numbers of crashes seen at the different types of roundabouts A, B1, B2, and B3 stem from selection bias and the improper constant rate assumption.

Higher traffic volumes, the presence of crosswalks, and painted bicycle lanes result in a higher number of crashes at B1 and B2 roundabouts. However, it does not justify the conclusion that in roundabouts, bicycles travelling in mixed traffic (Type A) have a reduced risk of crashing compared with types B1, B2, and B3, where bikes are travelling on dedicated infrastructure ([30]). When bicycles have right of way priority, the crash numbers are significantly lower, if the bicycle ford is not painted in red and if there is no zebra crossing. Both are features that indicate to the bicyclist that he or she has the right of way without question.

**Author Contributions:** Conceptualization, A.L. and P.W.; methodology, P.W. and A.L.; software, J.F., J.N. and G.S.; validation, P.W., A.L.; formal analysis, P.W. and J.N.; investigation, A.L. and P.W.; resources, A.L.; data curation, J.F.; writing—original draft preparation, A.L. and P.W.; writing—review and editing, A.L., G.S., J.N. and P.W.; visualization, A.L., J.F. and P.W.; supervision, P.W.; project administration, A.L.; funding acquisition, A.L. and P.W. All authors have read and agreed to the published version of the manuscript.

**Funding:** This research was funded by the German Ministry for Digital and Transport grant number 19F2082A. The APC was funded by the Publication Fonds of the German Aerospace Center.

**Institutional Review Board Statement:** Not applicable.

**Informed Consent Statement:** Not applicable.

**Data Availability Statement:** Not applicable.

**Conflicts of Interest:** The authors declare no conflict of interest.

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
