# Peer review of "Traffic Safety at German Roundabouts—A Replication Study"

_safety, 2011_

Round 1
Reviewer 1 Report
I am thankful for being given the opportunity to review the present research paper.
This is a generally well written paper describing a thoroughly done and interesting work.
Authors are advised to address some general comments.
For both researchers and practitioners, it would be useful to clarify whether the roundabouts of the current analysis were designed and constructed according to the German guidelines. Also, do roundabouts’ design and operational characteristics comply with the principles of Safe System? Have (any of) these roundabouts being road safety audited?
Authors are suggested to include information on functional classification for the intersecting roads and the speed environment for the different types of roundabouts of the current study.
Authors are advised to check the manuscript for any inconsistencies. For example, a check is needed with regard to the results of [1] and the safest roundabout type, in lines 31-32 on page 1 and line 190 on page 7.
Author Response
Dear Reviewer,
thank you taking time to review our paper.
We added in information on guideline conformity and a statement on functional classification because this, clearly, is important for interpreting the results of our study.
Also, it can make a lot of difference and introduce a bias, if a location was treated by an road accident commission or if there has been a road safety audit. While it is beyond the study to investigate on this for 1300 roundabouts, we have added the relevant information regarding these issues.
Thank you for revealing the inconsistencies. We have fixed the issues you have raised and tried to find more inconsistencies. We did not find more.
Best regards
Andreas Leich
Reviewer 2 Report
The paper presents titled “Traffic Safety at German Roundabouts – A Replication Study”, a replicate study of a work published in 2011. The paper seems to be appropriate for SAFETY journal. The manuscript is well organized, the quality of its text is good and the references list is appropriate.
However, it is difficult to state about the novelty of the paper, and it is not clear what the main findings of this study are.
I propose also these few changes:
1. The abstract should be fully understandable without reference to other sources. So, rewrite the abstract without citations.
As your work is replicate study, I think it is better to rewrite the abstract using author name (Bendizio) and presents the scope of his work and results. Then you may present the goal of your work, its focus, methodology and obtained results.
2. Using Abbreviations in scientific paper: as a rule, abbreviations and acronyms should be written out in full on their first use and then followed by the abbreviated form in parentheses. Faster-RCNN, CONV, DFGTS, RPN, NWSIB are some examples.
3. Scientific papers generally should be written in third person. Thus, rewrite the paper and avoid the use of "we" and "our." Using for example the passive voice sentences.
Author Response
Dear Reviewer,
thank you for your comments.
1. The abstract should be fully understandable without reference to other sources. So, rewrite the abstract without citations.
We have changed the abstract accordingly
2. Using Abbreviations in scientific paper: as a rule, abbreviations and acronyms should be written out in full on their first use and then followed by the abbreviated form in parentheses. Faster-RCNN, CONV, DFGTS, RPN, NWSIB are some examples.
Indeed. We have revised the paper and introduced every abbreviation.
3. Scientific papers generally should be written in third person. Thus, rewrite the paper and avoid the use of "we" and "our." Using for example the passive voice sentences.
We have changed to passive voice. Only statements that emphasize that it is the author's opinion and interpretation of results is "we" and "our"
Best regards
Andreas Leich
Reviewer 3 Report
The paper presents a combined study, by using both traditional and Machine Learning (ML)-based approach, to explore how different types of roundabouts can result in different levels of safety. In particular, the aim of the study was to expand the results and replicate some of the findings of a previous research (Bondzio, L.; Ortlepp, J.; Scheit, M.; Voß, H.; Weinert, R., 2011.). Furthermore, some additional conclusions have been drawn about the safety of bicyclists, not included in the original study. A minimal model has been also established with a small number of factors and acceptable prediction quality.
As said, the study starts from a previous one, that investigated the details of what renders roundabouts safe or dangerous, by classifying German roundabouts into four distinct classes named A, B1, B2, and B3 as a function of how bicycles travel respect to motorized traffic.
The results demonstrated that the crash-rate of the four classes differed considerably, with B3 (where bicycles have separate bike paths next to the main road) being the safest one; however, it was not completely clear which of the differences between the crash-rates displayed are statistically significant.
The present paper proposes a different approach: a number of data-bases regarding the German federal state Northrhine-Westfalia have been used and fused together to find a richer data-set. As a result, a set of about 1300 roundabouts was obtained; Machine learning (ML) was used to recognize traffic signs and find the traffic circles and their relevant features, even if the classification of roundabouts into A, B1 - B3 was performed manually.
The approach is very interesting and demonstrate the power of data processing and ML tools.
However, in the current presentation, the research appears very developed in terms of statistical analysis and treatment of data, but the identification of variables and factors that can affect the traffic safety (as a function of the geometric and functional characteristics of the road infrastructure and traffic conditions) seems very poor and coarse.
Also the interpretation and discussion of the results is too generic and based on single evidences, without actual opportunity to generalize it.
My suggestion is to better identify, in the very beginning of the article, the classification of roundabouts, and try to consider at least their main geometric features (diameter of the circle, number of legs, number of lanes, and so on). Afterwards, in the analysis of the results, it would be important to observe the influence of these factors respect to safety performances and traffic function.
The literature review can be extended, considering other research approaches to the problem of safety at road intersections, especially with regards to risk analysis for cyclists at roundabouts and consequent safety performances of different circular intersection layouts.
The organization of the paper need to be improved, by synthesising the text and restructuring the order of the sections in the first part (it would be important to firstly present the classification criteria); the scientific methodology is correct. The findings of the research are interesting but they can be more clearly presented.
In general, the paper in my opinion can be considered as an interesting approach to the use of data processing methods in order to investigate and better understand the investigated problem, but some improvements seems necessary.
Author Response
Dear Reviewer,
thank you taking time to review our paper.
You stated that the identification of variables and factors that can affect the traffic safety (as a function of the geometric and functional characteristics of the road infrastructure and traffic conditions) seems very poor and coarse.
During writing the paper we indeed were influenced by the perspective of Bondzio et al. and were undecisive on the level of detail we should provide without "copy pasting" their work. We have now revised our paper according to your suggestions.
My suggestion is to better identify, in the very beginning of the article, the classification of roundabouts, and try to consider at least their main geometric features (diameter of the circle, number of legs, number of lanes, and so on). Afterwards, in the analysis of the results, it would be important to observe the influence of these factors respect to safety performances and traffic function.
We followed your suggestion and revised the structure of our paper accordingly.
The literature review can be extended, considering other research approaches to the problem of safety at road intersections, especially with regards to risk analysis for cyclists at roundabouts and consequent safety performances of different circular intersection layouts.
Thanks for your recommendation. No list of references will be ever complete, our’s is no exemption. However, we have included some of the material outside there and put our results into relation to external material. We have pointed out differences and agreement between our results and the work cited in the references. So, it is our opinion that the current list is fine, and we do not know of additional references that are absolutely necessary to include. If the reviewer has a different idea, please let us know and we will be happy to extend our list.
Best regards
Peter Wagner and Andreas Leich
Round 2
Reviewer 1 Report
The authors have sufficiently addressed my comments. Overall, the paper has been improved considerably.
Reviewer 3 Report
No additional comments and suggestions for Authors.